# QSAR-Based Drug Repurposing and RNA-Seq Metabolic Networks Highlight Treatment Opportunities for Hepatocellular Carcinoma Through Pyrimidine Starvation

**DOI:** 10.3390/cancers17050903

**Published:** 2025-03-06

**Authors:** Nicholas Dale D. Talubo, Emery Wayne B. Dela Cruz, Peter Matthew Paul T. Fowler, Po-Wei Tsai, Lemmuel L. Tayo

**Affiliations:** 1School of Chemical, Biological, and Materials Engineering and Sciences, Mapúa University, Manila 1002, Philippines; nddtalubo@mymail.mapua.edu.ph (N.D.D.T.); ewbdelacruz@mymail.mapua.edu.ph (E.W.B.D.C.); pmptfowler@mapua.edu.ph (P.M.P.T.F.); 2School of Graduate Studies, Mapúa University, Manila 1002, Philippines; 3Department of Biology, School of Health Sciences, Mapúa University, Makati 1203, Philippines; 4Department of Food Science, National Taiwan Ocean University, Keelung 202, Taiwan; powei@mail.ntou.edu.tw

**Keywords:** drug repurposing, QSAR, hepatocellular carcinoma, pyrimidine metabolism

## Abstract

The challenges associated with the efficacy of systemic therapy options for Hepatocellular Carcinoma (HCC) are contributing to the high mortality rate of the disease in advanced cases. There is an urgent need to develop new drugs and treatment regimens that are effective regardless of HCC’s molecular heterogeneity. However, the costly and uncertain nature of drug development often deters risk-averse investors. To address this, the study employs a combination of drug repositioning and metabolic vulnerability searching to identify approved drugs with high potential to inhibit an essential metabolic pathway. Overall, this research highlights the metabolic vulnerabilities of HCC and proposes the testing of the suggested approved drugs as potential anti-HCC therapies.

## 1. Introduction

Cancer therapeutics is a growing field with multiple avenues being developed. In 2018–2022, or five-years’ worth of cancer case statistics, the annual cancer death rate is still approximately 146 per 100,000 cancer patients. Some cancers are disproportionally more lethal than others, and liver cancer is among the top seven most lethal cancers in both men and women [1]. In terms of therapeutics, common options for cancer include surgery, radiotherapy, and chemotherapy. However, the success of these treatments is varied and is known to depend on the site, type, and stage of cancer, along with the patient’s response to the treatment [2].

For hepatocellular carcinoma (HCC), the most common type of liver cancer, treatment options are guided by the widely accepted Barcelona Clinic Liver Cancer (BCLC) algorithm. This treatment strategy relies on the characterization of HCC into five stages primarily determined by prognostic markers. These markers include tumor burden, liver function, and physical status, which are further refined by Alpha-Fetoprotein (AFP), Albumin–Bilirubin Index (ALBI) score, Child–Pugh, and Model for End-Stage Liver Disease (MELD) [3]. Afterwards, individualized approaches are decided based on the prognosis group of the patients and the presentation of their HCC.

The BCLC system suggests a specific treatment out of the three currently existing options for the first cancer treatment. For those in the early stages, ablation, resection, and transplantation are the primary treatment options. A failed treatment response or a tumor detected at the intermediate stage would need Transarterial Chemoembolization (TACE), which if successful, will downstage the tumor and allow successful transplantation for those without contraindications. However, if the tumor is advanced, a three-line systemic treatment is proposed, using monoclonal antibody and/or tyrosine kinase inhibitor (TKI) medication [3].

Research into effective standards drugs for unresectable HCC is still an active field. Recently, the drug combinations atezolizumab/bevacizumab and tremelimumab/durvalumab have been shown to be alternatives that can replace sorafenib and circumvent its drawbacks of toxicity and bioavailability [4,5]. However, challenges remain in the development of effective systemic treatments. The major drivers of HCC are currently undruggable, and research has yet to confirm alternative highly mutated targets. This issue is further compounded by the lack of biomarkers capable of predicting treatment efficacy. For instance, immunotherapy research for HCC requires biomarkers beyond PD-L1, microsatellite instability, and tumor mutational burden (TMB), as these have limited evidence of effectiveness in predicting immunotherapy responses [6].

To spearhead the identification of possible biomarkers, computational tools like weighted correlation network analysis (WGCNA) are employed, which are fueled by the public curation of cancer omics data in databases like The Cancer Genome Atlas (TCGA) and Gene Expression Omnibus (GEO) [7,8]. For HCC, the application of these methods can be complicated by the cancer’s heterogeneity, as different molecular mechanisms may exist for different etiologies [9]. The identification of biomarkers for predictive and treatment purposes is crucial for both the development of new treatments and the evaluation of their efficacy, as well as that of existing treatments.

In this paper, potential therapeutic targets in HCC are determined using metabolic modeling, focusing on targets predicted to disrupt tumor metabolism and cause fatal outcomes upon knockdown. These targets must demonstrate consistent disruption across different molecular subtypes of HCC. To facilitate systematic treatment development, the study incorporates drug repurposing by employing QSAR analysis of known inhibitors for the selected genes, with molecular docking simulations serving as confirmation. This paper then serves as a framework for future in vitro studies.

## 2. Materials and Methods

### 2.1. Transcriptomic Dataset Retrieval

The R library TCGAbiolinks version 2.32.0 was utilized to retrieve RNA-seq data [10,11]. The GDCquery function was employed to obtain the raw data for the TCGA-LIHC cohort. Subsequently, the GDCprepare function was used to generate a SummarizedExperiment object, from which the metadata and count matrix were extracted. Finally, to ensure data heterogeneity, PanCancerAtlas subtype information was retrieved and used to filter the samples so that only those with subtype information would be used for the analysis [12]. The expression data underwent further preprocessing for analysis using gmctool. A gene list obtained from Human1 was applied to exclude unused genes, while genes with a median expression of less than five or those not expressed were removed. Additionally, only complete cases were retained, and the analysis was restricted to samples classified as primary solid tumor. As a result, 3629 genes, three subtypes, and 160 samples were included in the subsequent analyses.

### 2.2. Gene Essentiality Analysis

To identify potential metabolic target genes, an automated approach was employed using the R application gmctool (version 1.0.0). As detailed in its accompanying publication, gmctool leverages Genetic Minimal Cut Sets (gMCSs) and cancer RNA-seq data to predict metabolic vulnerabilities that could be lethal to cancer cells. gMCSs do this through perturbation such as mutations, deletions, or any type of alterations that when knocked out make a metabolic task impossible to do. Through this, it finds applications in cancer research in finding key biomarkers that can be utilized in finding novel ways of dealing with cancer with more specificity and less side effects [13]. All essential metabolic tasks were considered during the search for knockout (KO) genes, and simulated biomass production restrictions were applied. The developed gmcsTHX gene thresholding method was selected with a percentile expression threshold of 0.05. Genes with expression levels above this threshold were classified as “highly expressed” (ON), while those below the threshold were labeled as “lowly expressed” (OFF). Using this threshold, SingleKO and DoubleKO gene candidates were calculated and exported for further analysis.

### 2.3. Functional Enrichment of Lethal Genes

The exported SingleKO genes were filtered to include only those classified as KO genes across all samples within each subtype. These selected genes were then subjected to Overrepresentation Analysis (ORA) using the enrichKEGG function from the R package clusterProfiler (version 4.12.6) [14]. Subsequently, the enrichment results were refined using KEGG Pathway database categories to retain only pathways within the Metabolism category. Additionally, the protein–protein interaction (PPI) network of the selected genes was constructed using STRING (version 12.0) [15]. A confidence score threshold was set at 0.7 and relationships based on text mining alone were removed from the network. The selected active interaction sources include Experiments, Databases, Co-expression, Neighborhood, Gene Fusion, and Co-occurrence. Furthermore, the Markov Clustering Algorithm was applied.

Genes involved in DoubleKO interactions were filtered to include only those present in the SingleKO gene list. Moreover, the DoubleKO interaction data defined as essential pairs generated by gmctool were integrated into the STRING network and visualized using a gene interaction matrix. In summary, the genes of interest were selected based on their classification as SingleKO genes and as essential pair DoubleKO interactions.

### 2.4. Identification and Comparison of Differentially Expressed Genes

The main transcriptomic dataset was further filtered to include only samples with available adjacent normal tissues. Differential gene expression analysis was conducted using DESeq2 (version 1.44.0), with gender and tissue type included in the design matrix. The contrast was set to reference “Solid Tissue Normal” samples, and the results were extracted. A *p*-value threshold of 0.05 and a log2FC cutoff of 1.0 were applied. A volcano plot was generated using ggplot2 (version 3.5.1), plotting the −log_10_(*p*-value) against the fold change.

Using the prcomp function, the principal components of the identified differentially expressed genes (DEGs) were calculated. The data were centered and scaled within prcomp. The first two principal components (PC1 and PC2) were used to create a PCA plot, with points colored by sample type. Lastly, the DEGs were translated into EntrezID using org.Hs.eg.db and intersected with the SingleKO genes.

### 2.5. External Validation

To further confirm the validity of the identified genes, an external dataset was utilized. The external dataset was retrieved from recount3 (version 1.14.0) using the project ID SRP174991, with the organism specified as human, annotation as gencode_v26, and type as genes. The dataset consisted of 70 samples, including 35 HCC samples paired with 35 normal liver tissue samples.

The external dataset was preprocessed as similarly as possible to the main dataset. For unsupervised clustering, the variance-stabilizing transformation from DESeq2 was applied. Genes were filtered to include only those present in the SingleKO gene list from the main dataset. Given the a priori knowledge of the number of clusters, k-means clustering was performed with nstart = 25 and two centers. A heatmap was generated using ComplexHeatmap (version 2.20.0). A contingency table was created between the actual tissue type and the computationally identified clusters. Fisher’s exact test was used with the fisher.test function to assess the association of the two classifiers. Finally, the dataset was processed through gmctool using the same parameters as those for the original dataset. Similar to the previous analysis, only genes that proved to be lethal upon knockout across all available HCC samples were considered and compared with the original list.

### 2.6. ChEMBL Data Collection

The Entrez IDs of the selected genes of interest were converted to their UniProt IDs using the ID mapping service of UniProt (accessed 25 December 2024). These UniProt IDs were subsequently mapped to their corresponding ChEMBL IDs using ChEMBLdb release 35. The IC50 data for compounds tested against the selected proteins were retrieved from the same ChEMBL release. These proteins were further filtered based on the availability of compound data. DHODH and TYMS were ultimately selected for further analyses.

The ChEMBL database was queried using the Python library chembl_webresource_client (version 0.10.9) [16]. The retrieved data for the two proteins underwent several preprocessing steps prior to their use in QSAR modeling. First, compounds with non-numerical IC50 values or missing units were removed from the dataset. Next, all remaining units were standardized to molarity and converted to pIC50 using Equation (1). The Canonical SMILES of the remaining compounds were then queried using the same database. Furthermore, using the Canonical SMILES of the selected compounds, chemical clustering was conducted with their atom pairs using ChemmineR (version 3.58.0).pIC50 = −log_10_(IC50)(1)

### 2.7. Flux Balance Analysis and Single Gene Knockout

To further confirm the essentiality of the selected genes and their effects on metabolism, flux balance analysis (FBA) was conducted using Human-GEM retrieved from https://github.com/SysBioChalmers/Human-GEM (accessed on 27 February 2025), a comprehensive metabolic model for human cells. The model was imported into the software using the COBRApy library (version 0.29.1). The medium was configured by retaining only the fluxes of transport reactions that approximate metabolites present in human blood, as detailed in Appendix A. Additionally, the minimal amount of medium required to support growth was computed using slim_optimize and set as the available medium. Flux balance analysis was performed three times: once under normal conditions and twice under single-gene knockout conditions for DHODH and TYMS. Key flux information was extracted from the solution space, including fluxes associated with specific reactions of interest, such as DNA synthesis (MAR07160), RNA synthesis (MAR07161), dATP (MAR07861), dCTP (MAR07860), dGTP (MAR07862), dTTP (MAR07857), UTP (MAR07796), and biomass (MAR13082).

### 2.8. D Structure Generation and Molecular Descriptors Calculation

Using the queried SMILES, the 3D structures of each compound were generated with OpenBabel (version 3.1.1-8). The gen3d operation was employed with the “best” option enabled to ensure improved force field optimization and a larger conformer search space [17]. Subsequently, each generated 3D structure was imported into Python (version 3.13.1) as a mol object using the RDKit library (version 2024.09.06). To compute 3D descriptors, the Mordred Community Python library (version 2.0.6) was utilized, with the ignore_3D option set to false [18]. The community-maintained version of Mordred was specifically chosen to avoid compatibility issues with more recent Python versions (https://github.com/JacksonBurns/mordred-community) (accessed on 18 December 2024). A total of 1756 3D descriptors were calculated. To ensure consistency, the same procedures were later applied to the molecules retrieved from DrugBank.

### 2.9. Feature Selection and QSAR Model Training

To reduce the number of features, the dataset underwent feature selection across all three models. To ensure consistency, Sequential Feature Selection (SFS) was applied to each model. Initially, both datasets were subjected to low variance filtering to remove uninformative features. Subsequently, features with high inter-correlation and low correlation with pIC50 were eliminated. Using scikit-learn’s StandardScaler, each feature was standardized to have a mean of zero and a standard deviation of one [19]. Following this preprocessing, the top 100 optimal features per model were selected using SFS in the forward direction and scored using R^2^, with a cross-validation setting of 5.

Three algorithms were chosen to build the models, two of which were recommended by a benchmarking study. These algorithms included Support Vector Regression (SVR), Ridge Regression, and XGBoost Regression. Each algorithm represents a distinct domain of machine learning: Support Vector Machines, linear models, and decision trees, respectively. The dataset was split randomly, with 80% used for training and 20% reserved for testing. An iterative strategy was employed to identify the optimal training parameters, ranking model performance based on metrics such as MAE, RMSE, R^2^, and PCC. The calculation of these metrics and model training were partially informed by publicly available code from a QSAR-based drug repurposing study available in https://github.com/manojk-imtech/anti-HCV (accessed on 15 December 2024) [20]. Finally, the best-performing model was selected for pIC50 prediction.

### 2.10. DrugBank Database Collection

DrugBank 6.0 was utilized to define the chemical space for identifying inhibitors of the two selected proteins from the final filtering with compound data [21]. To narrow the search space, only approved drugs were considered for repositioning. The 3D structures of the approved compounds were retrieved from DrugBank and imported into Python (version 3.13.1). Molecular descriptors of the compounds were predicted using Mordred, following the same procedures applied to the training and test data. To evaluate the applicability of the QSAR model, only compounds with a Tanimoto similarity greater than 0.5 to the nearest compound in the training set were considered for prediction [22]. The generic names of the drugs were subsequently extracted using the available SDF files.

### 2.11. Molecular Docking

To confirm that the selected compounds have the potential to dock to the proteins of the genes of interest, AutoDock Vina version 1.2.6-1 was used for molecular docking [23]. The protein structures for DHODH and TYMS were retrieved from the RCSB Protein Data Bank. For DHODH, structure 6J3B was used for molecular docking [24]. The structure was cleaned using PyMOL version 3.0.0 and the Prepare Receptor tool in usegalaxy.eu (accessed on 1 January 2025). The binding site for the repositioned drug was assumed to be the same as that of the inhibitor in the structure; thus, its centroid was determined, and a bounding box was defined accordingly.

A similar procedure was applied to TYMS. However, the structure with the inhibitor raltitrexed (5X5Q) had a resolution higher than 2.5 Å. To address this, another structure (5X5D) from the same study was used, importing the computed centroid of the binding site from 5X5Q [25]. A bounding box with a size of 20 units along the *x*, *y*, and *z* axes was defined, and an exhaustiveness of 25 was set. The most favorable conformation of the ligand was selected and further analyzed for interactions using BIOVIA Discovery Studio 2023.

## 3. Results

The query used for TCGAbiolinks successfully retrieved the TCGA-LIHC cohort and the PanCancerAtlas subtypes of HCC based on the official paper released by The Cancer Genome Atlas Research Network [12]. The paper successfully identified three subtypes influencing patient survivability and molecular presentation. The transcriptomic data were annotated with the corresponding subtype information, dropping those that lacked it or had incomplete data. A total of 160 samples were kept, with 57 belonging to iCluster1, 49 belonging to iCluster2, and 54 belonging to iCluster3.

Metabolic modeling was performed with gmctool to determine lethal gene knockout opportunities. The output knockout genes were filtered to include only those that were knocked out in all of the samples per subtype, indicating its importance regardless of molecular presentation. The raw data for the single knockout gene results from gmctool are presented in Appendix A. A total of 278 genes were found to be lethal to HCC cells when knocked down. To assess the biological function of these genes, overrepresentation analysis was conducted as seen in Figure 1.

The filtered gene list appears to be involved in various metabolic functions that are potentially significant to HCC. Among the pathways identified, oxidative phosphorylation demonstrated the highest enrichment factor and gene count compared to other metabolic terms, underscoring its importance. Additionally, pathways related to lipid, amino acid, and nucleotide metabolism were prominently represented in the results. These pathways encompass both broad categories and more specific metabolic processes.

To examine the potential significance of the metabolic genes of interests, differential gene expression analysis was conducted on samples with available paired adjacent normal liver tissue. The results of the analysis can be seen in Figure 2.

As expected, Figure 2 highlights the expression differences between normal and solid tumor samples used in the gene essentiality analysis conducted with gmctool. The observed separation of the samples in the PCA plot (Figure 2B) further confirms this distinction. Additionally, the potential heterogeneity of the HCC samples is evident, as some data points appear to lie farther from the centroids of their respective sample types.

The list of differentially expressed genes (DEGs) was compared with SingleKO genes of interest, revealing that 37 genes were both differentially expressed and identified as potential single knockout targets. These genes are *ACAA1*, *CAD*, *CYP26A1*, *GNPAT1*, *DHODH*, *SQLE*, *EHHADH*, *ECHS1*, *BBOX1*, *COX4I2*, *GCH1*, *PSAT1*, *EPRS1*, *CYP1A2*, *MMAA*, *COX6A2*, *FLAD1*, *ALB*, *ACAA2*, *DTYMK*, *FASN*, *COX7B2*, *RRM2*, *SPTLC3*, *TYMS*, *ACADSB*, *ND6*, *CYTB*, *ND2*, *ND5*, *COX1*, *ND3*, *ND4*, *ND1*, *ATP6*, *ND4L*, and *ATP8*. The complete list of DEGs is provided in Appendix A.

To account for potential biases in the dataset, the SingleKO metabolic gene list was used for unsupervised clustering of an independent external dataset comprising HCC and normal liver tissue samples. The same preprocessing steps and transformations were applied to this dataset. Using only the gene list identified in this study as features, two clusters were generated using k-means clustering and statistically compared to the actual status of the samples. The results are shown in Figure 3.

Cluster 1 appeared to be associated with HCC samples, while Cluster 2 was associated with normal samples. The association between the computationally identified clusters and the actual tissue classifications was tested using Fisher’s Exact Test, yielding a *p*-value of 4.481 × 10^−12^. This strongly indicates a significant association between the actual tissue groups and the identified clusters. The result is further supported by the high odds ratio of 86.9054, with a large effect size (95% confidence interval: 15.48 to 952.82) that does not include one.

The external dataset was also processed using gmctool to assess the reproducibility of the identified gene list. The same filtering parameters were applied to the SingleKO results of the external dataset to generate a gene knockout list computed to be effective across all samples in the dataset. Interestingly, 275 overlapping genes were identified between the two SingleKO lists, further supporting the importance of the identified genes. A Euler plot visualizing the relationship is provided as Appendix A. Furthermore, the intersecting genes were placed in Appendix A.

To further validate the biological significance of the main gene list, its Protein-Protein Interaction (PPI) network was generated using STRINGdb. Figure 4 illustrates the PPI network of the gene list. A highly significant PPI enrichment *p*-value (<1.0 × 10^−16^) was observed, indicating strong interactions among the proteins. With a high confidence score threshold of 0.7 to 0.9, the network contained a total of 3386 edges, substantially exceeding the expected number of 414 edges. While the network is highly connected, fragmentation was observed, potentially constraining topological analysis of the network. To further contextualize the PPI, the Markov Cluster Algorithm was used to determine protein clusters. Overall, a high agreement between dense clusters and their functional pathways can be observed. The results of the STRINGdb analysis are provided in Appendix A.

Double knockout (DoubleKO) genes were subsequently evaluated as potential therapeutic targets. gmctool is designed to calculate both single knockout (SingleKO) and DoubleKO interactions based on input RNA-seq data. The results of the analysis include information on essential pairs; these gene pairs are SingleKO genes and are potential DoubleKO interactions identified through gMCS. Figure 5 depicts the gene interaction matrix with the essential pairs interactions identified by gmctool.

As observed above, the majority of genes lack annotated relationships. This is expected, as heatmaps require the examination of one gene against all others, which amplifies the prevalence of the “no connection” relationship type. Following this, the next most common relationships are those derived from STRINGdb. In contrast, the potential double knockout interactions classified as essential pairs by gmctool account for only roughly half of the STRINGdb relationships in the matrix. Lastly, the overlap between essential pairs and STRINGdb relationships has the lowest representation. While this low representation may seem concerning, it is important to note that the essential pairs identified by gmctool represent the minimal set of genes whose simultaneous knockdown would disrupt the metabolic pathway. As such, their relationships may not be captured by direct connections.

Genes that are potentially involved in DoubleKO interactions that were also classified as SingleKO genes were identified as the final potential targets for QSAR modeling. Additionally, all pathways containing these genes were included in the database search. After applying the specified criteria, a total of 85 unique genes were identified. Among these, 19 ChEMBL IDs were associated with proteins encoded by the genes. Some of the genes appear to be represented only by one ChEMBL ID with the classification of multiple proteins. By querying the ChEMBL database for IC50 data, two proteins with an adequate number of unique compounds and IC50 data were identified: DHODH (CHEMBL1966) and TYMS (CHEMBL1952). These proteins had 1581 and 391 compounds available, respectively, for QSAR modeling. These compounds were clustered to examine the diversity of the compound library. Appendix A contains the 3D clustering results for DHODH and TYMS compounds.

Since the analysis has now focused on DHODH and TYMS, it is important to contextualize their roles. As depicted in the STRINGdb network (Figure 4) and its MCL clustering results (Appendix A), DHODH and TYMS remain interconnected within the protein-protein interaction (PPI) network. They were successfully assigned to Cluster 23 and Cluster 7, respectively, with these clusters associated with pyrimidine biosynthesis, 2-deoxyribonucleotide biosynthesis, and the interconversion of nucleotide di- and triphosphates. This clustering is further supported by their KEGG pathway membership, as revealed by the enrichment analysis in Figure 1, and by the literature reviewed in the Discussion section. Specifically, DHODH was uniquely linked to the metabolic pathway “Biosynthesis of Cofactors”, while TYMS was associated with the “One-Carbon Pool by Folate” pathway. Both proteins, however, play critical roles in pyrimidine metabolism, which is part of the broader nucleotide metabolism pathway.

To evaluate their potential impact on nucleotide metabolism and the overall growth of cancer cells, flux balance analysis (FBA) was conducted. Given the focus on understanding their roles in metabolism, a stoichiometrically balanced metabolic network was required. For this purpose, Human-GEM was employed as the metabolic model. Appendix A shows the growth media used for modeling, while Appendix A provides an overview of only the primidine metabolism subsystem of the model. Using flux balance analysis (FBA), the knockdown of the genes DHODH and TYMS was simulated for the whole model. The resulting effects on their fluxes, compared to the no-knockdown scenario, are presented in Figure 6 and Table 1.

The knockdown of DHODH and TYMS significantly altered the flux balances of the model. Since the objective function was set to maximize biomass, it can be inferred that, if the knockdown of both genes decreased the flux significantly, it has the potential to be detrimental to achieving optimal growth. DHODH, in particular, is associated with multiple functions, necessitating further assessment of its effects on nucleotide metabolism. To evaluate these effects, the fluxes associated with the artificial reactions of DNA and RNA synthesis, as well as the fluxes of nucleotides used for DNA and RNA synthesis, were extracted from the solution space.

As shown in the results, the knockdown of DHODH caused a pronounced decrease in the availability of key nucleotides. For instance, the fluxes of dATP, dCTP, dGTP, dTTP, and UTP dropped dramatically, with values decreasing from 3.74 × 10^−1^, 2.49 × 10^−1^, 2.49 × 10^−1^, 3.74 × 10^−1^, and 3.10 × 10^1^ mmol/gDW/h (normal conditions) to 3.22 × 10^−15^, 2.14 × 10^−15^, 2.14 × 10^−15^, 3.22 × 10^−15^, and 8.12 × 10^−15^ mmol/gDW/h, respectively. This reduction in nucleotide availability translated to a significant decline in biomass production, from 4.67 × 10^1^ mmol/gDW/h (normal) to 4.02 × 10^−13^ mmol/gDW/h (DHODH knockout). Similarly, the knockdown of TYMS resulted in a dramatic decrease in fluxes, with values approaching zero across all measured parameters. For example, DNA synthesis dropped from 1.25 × 10^0^ mmol/gDW/h (normal) to 1.91 × 10^−13^ mmol/gDW/h (TYMS knockout), and RNA synthesis decreased from 1.72 × 10^2^ mmol/gDW/h (normal) to 8.05 × 10^−13^ mmol/gDW/h (TYMS knockout). The results emphasize their potential as enzymes to control the metabolic pathway.

The molecular descriptors of each compound were computed with Mordred. A total of 1827 descriptors were computed including both 3D and 2D. These descriptors were cleaned, preprocessed, and reduced to their 100 most important features. These features were used to train QSAR models where multiple parameters were iteratively tested. Table 2 summarizes the highest quality models produced per algorithm.

Three different machine learning algorithms were employed to train the QSAR data. For DHODH, the R^2^ values ranged from 0.7679 to 0.8210. The SVM algorithm achieved the highest R^2^ at 0.8210, followed by the XGBoost algorithm at 0.7679 and the Ridge Regression algorithm at 0.7667. SVM also exhibited lower MAE and RMSE compared to the other models. However, the Ridge Regression model showed a slightly better MAE than the XGBoost model. The PCC values, reflecting the correlation between predicted and actual pIC50 values, followed the order: SVM > XGBoost > Ridge Regression. Based on all evaluation metrics, it was concluded that SVM performed the best on unseen test data for DHODH.

For TYMS, the R^2^ values ranged from 0.6022 to 0.8101, with the same ranking observed: SVM achieved the highest R^2^ at 0.8101, followed by XGBoost at 0.7638, and Ridge Regression at 0.6022. Unlike the results for DHODH, a consistent pattern was observed across MAE and RMSE, where Ridge Regression performed the worst, with values of 1 for both metrics. Ultimately, all evaluation metrics confirmed that SVM outperformed the other models on unseen test data for TYMS. This can be further confirmed with the predicted vs. actual plots of the test data in Figure 7. The parameters used and the resulting performance of the models were placed in Appendix A.

Using the best-performing models, the pIC50 values of drugs annotated as approved on DrugBank were predicted for DHODH and TYMS. First, the top 20 pIC50 values for DHODH were identified by applying the SVM model to the molecular descriptors of approved drugs. Each drug was then docked to the active site of DHODH, determined based on the known binding site of Teriflunomide, a recognized inhibitor. The compound with the most negative binding energy for each docking was selected. Additionally, only drugs with a Tanimoto similarity greater than 0.5 to a compound in the training set were considered, ensuring the applicability of the QSAR model. Figure 8 shows the density of considered drugs in the approved drugs of the DrugBank database. Finally, the indications of each drug were reviewed using the DrugBank database, and their relevance to cancer and HCC was noted. The collected data can be seen in Table 3.

From the top 20 drugs with potential to inhibit the target, the top three compounds with the most favorable binding energies that have not yet been used for cancer or HCC were selected. First, Oteseconazole (DB13055) had a binding energy of −12 kcal/mol, a predicted pIC50 of 7.5823, and a nearest compound similarity of 0.59. Next, Tipranavir (DB00932) exhibited a binding energy of −11.4 kcal/mol, a predicted pIC50 of 7.3318, and a Tanimoto similarity to the nearest compound in the training set of 0.56. Finally, Lusutrombopag (DB13125) demonstrated a binding energy of −11 kcal/mol, a predicted pIC50 of 7.3279, and a Tanimoto similarity to the nearest compound in the training set of 0.56. These three compounds were compared to Teriflunomide (DB00255), which had a binding energy of −9.8 kcal/mol and a predicted pIC50 of 6.3896 compared to the actual pIC50 retrieved from ChEMBL at 6.3767. These indicate that the three compounds have a high potential to bind to DHODH and inhibit it.

The top three selected drugs were docked with DHODH at the binding site of Teriflunomide. Figure 9 visualizes the resulting protein–ligand complexes and their interactions. As Figure 9 shows, the three compounds exhibited similar interactions to those of the Teriflunomide control. However, Oteseconazole displayed additional pi–sigma, pi–sulfur, and pi–pi stacked interactions. Notably, the pi–sigma interaction was also observed in Tipranavir and Lusutrombopag.

Similarly, the top 20 drugs with the potential to inhibit TYMS were considered. As Table 4 shows, the highest binding energy meeting the set criteria was observed for Tadalafil (DB00820), with a binding energy of −9.9 kcal/mol, a predicted pIC50 of 7.5070, and a Tanimoto similarity to the nearest compound of 0.61. This was followed by Dabigatran (DB14726), which had a binding energy of −9.5 kcal/mol, a predicted pIC50 of 7.2764, and a Tanimoto similarity of 0.54.

Two compounds shared the same binding energy of −9.3 kcal/mol. Baloxavir marboxil (DB08903) exhibited a predicted pIC50 of 7.3658, while Candesartan cilexetil (DB00796) showed a predicted pIC50 of 7.2675. Although their predicted pIC50 values were similar, their Tanimoto similarity to the nearest compound differed, at 0.70 and 0.59, respectively.

Additionally, Raltitrexed, a known TYMS inhibitor, was docked and found to have a predicted pIC50 of 6.8772, compared to its actual pIC50 of 6.0555. Similar to the results for DHODH, the four selected compounds displayed higher binding energies and significant potential to inhibit TYMS.

Figure 10 illustrates the resulting protein–ligand complexes and their interactions with TYMS. As Figure 10 shows, the Raltitrexed control exhibited van der Waals, conventional hydrogen bond, carbon–hydrogen bond, pi–cation, pi–donor hydrogen bond, pi–pi T-shaped, and pi–alkyl interactions. Similar interactions were observed for Tadalafil, with the addition of an alkyl interaction. In contrast, Dabigatran displayed a pi–pi stacked interaction but lacked the pi–pi T-shaped interaction seen in the control. The compound Baloxavir marboxil demonstrated unique interactions, including halogen (fluorine), pi-anion, and alkyl interactions. Finally, Candesartan cilexetil exhibited similar interactions as the other compounds but also displayed an unfavorable donor–donor interaction.

## 4. Discussion

The failure of first-line treatments such as ablation, resection, and transplantation for HCC patients often necessitates systemic therapy as a last resort. While these first-line treatments are generally effective, most cases of HCC are diagnosed at late stages, leaving systemic therapy as the only viable option for many patients [26]. This heavy reliance on systemic therapy in mid-to-late-stage HCC patients is concerning, as high rates of acquired drug resistance are commonly associated with the disease [27]. Various mechanisms of drug resistance have been identified in HCC, depending on the resistance factor. One such mechanism is metabolic reprogramming.

HCC, like all cancers, requires metabolic reprogramming to meet its increasing energy and material demands. This reprogramming not only impacts the effectiveness of drug treatments but is also critical for the survival of HCC cells [28]. Similar to many cancers, the Warburg effect is observed in HCC, indicating a shift from oxidative phosphorylation (OXPHOS) to glycolysis to fulfill energy requirements. However, recent studies have shown that HCC exhibits metabolic flexibility, switching between OXPHOS and glycolysis based on glucose availability and hypoxic stress [29].

This adaptability suggests that disrupting the function of both pathways is essential for depriving HCC cells of energy. Interestingly, metabolic modeling of HCC cases in this study revealed a higher number of essential genes and synthetic lethality opportunities in the OXPHOS pathway compared to glycolysis. This uneven distribution may pose a challenge in developing drugs capable of simultaneously targeting both pathways to effectively induce energy deprivation in HCC cells.

Targeting other essential metabolic pathways is crucial to expanding the tools available for the systemic treatment of advanced HCC. However, both drug discovery and repurposing are constrained by the availability of compound data and its biological effects. Therefore, it is vital to ensure that, when targeting specific genes, a diverse set of compound data are available and that the potential for their essentiality is high. Through proper filtering, this study identifies two essential genes, DHODH and TYMS, whose individual knockdown could cause aberrant effects in the pyrimidine metabolism of HCC.

Pyrimidine metabolism is composed of three essential interrelated pathways: de novo synthesis, the salvage pathway, and the catalytic pathway [30]. In normal mature cells, the salvage pathway serves as the primary source of pyrimidines. However, in cancer cells, their proliferative nature necessitates a constant reliance on the de novo synthesis pathway, which provides a continuous supply of deoxyribonucleoside triphosphates (dNTPs) [31]. Additionally, a recent study has highlighted the role of pyrimidine metabolism in facilitating and maintaining the epithelial–mesenchymal transition (EMT) in HCC [32]. This process significantly enhances the metastatic potential of cancer cells and contributes to drug resistance.

The genes DHODH and TYMS play crucial roles in the de novo pyrimidine metabolism pathway, acting as enzymes in the fourth and tenth steps, respectively. DHODH catalyzes the conversion of dihydroorotate into orotate, a key intermediate required for the production of deoxythymidine monophosphate (dTMP) through subsequent processing [33]. Inhibition of DHODH has been associated with anti-proliferative effects, reduced metastasis, and increased apoptosis in various cancers [34,35,36]. Since DHODH is a mitochondrial enzyme involved in the antioxidant pathway, its inhibition has been noted to increase the sensitivity of HCC to ferroptosis [37]. Moreover, compared to other cancers, HCC exhibits the highest mRNA expression levels of DHODH [33].

TYMS plays a central role in the de novo pyrimidine metabolism pathway by methylating deoxyuridine monophosphate (dUMP) to produce deoxythymidine monophosphate (dTMP) [38]. Knockdown of TYMS has been shown to increase apoptosis in certain cancers and negatively affect the metastatic potential of advanced cancers, including HCC [39,40,41]. In HCC tissues and cell lines, TYMS knockdown has demonstrated anti-proliferative effects and has been reported to reduce cancer aggressiveness via the TS/DYPD axis [31].

However, the suppression of TYMS alone may produce effects that depend on the tumor’s epithelial–mesenchymal transition (EMT) state. For tumors that have completed EMT, one study observed that TYMS knockdown could potentially increase metastasis, as cancer cells in a partial EMT state are known to exhibit a higher metastatic potential [42].

Figure 11 summarizes the de novo pyrimidine synthesis pathway. The results of this study highlight the essentiality of DHODH and TYMS as single-gene targets to suppress HCC proliferation. As Figure 11 shows, the suppression of DHODH can potentially cause downstream effects for the production of cytosine, uracil, and thymine. However, it is important to note that suppressing DHODH alone may allow pyrimidine synthesis to persist via the salvage and catalytic pathways, albeit to a limited extent. While inhibiting an enzyme in the salvage pathway could address this issue, its critical role in normal cells could render such treatment toxic. By simultaneously targeting TYMS and DHODH, a synergistic effect could be achieved, potentially leading to pyrimidine starvation, reversal of EMT, and a secondary effect of ferroptosis without compromising the salvage and catalytic pathways that are essential for the proper functioning of normal mature cells.

This is further supported by the results of the FBA and the subsequent simulated fluxome, wherein the biomass flux approached zero when single knockouts of DHODH and TYMS were simulated. The effects on nucleotide metabolism were also approximated, showing a decrease in DNA and RNA synthesis, along with a corresponding reduction in the availability of nucleotides required for their synthesis. However, since biomass was set as the objective function, the flux values were potentially limited by the mass-balancing constraints of the method and model. This suggests that the observed decrease in purine precursors flux may be a direct consequence of the mass-balancing requirements inherent to the method. Consequently, the effects on pyrimidine metabolism are more strongly supported by the literature cited above, the association of the genes with the pyrimidine metabolism pathway, and the context provided by the PPI network from STRINGdb.

Through QSAR modeling, this study identifies a variety of compounds capable of inhibiting one of the target genes. As Table 3 and Table 4 show, the application of the best-performing QSAR model to approved drugs in DrugBank revealed a set of promising compounds with higher predicted pIC50 values compared to the known inhibitors of TYMS and DHODH. Interestingly, several of the compounds with higher predicted pIC50 values are already used in cancer treatment as inhibitors of kinase enzymes. This aligns with findings from other drug-repurposing studies on DHODH and TYMS, which have also highlighted the potential of kinase inhibitors, commonly referred to as “nibs”, for targeting these genes [43,44,45].

The compounds with the most promising binding energy were selected for 2D protein–ligand interaction plotting and further analyses. For DHODH, the highest binding affinity was achieved by Oteseconazole. Oteseconazole is a novel tetrazole antifungal drug belonging to the therapeutic category of azole antifungals and is indicated for the treatment of recurrent vulvovaginal candidiasis (RVVC). While the drug has not yet been tested for anti-proliferative effects, it is noteworthy that azole compounds have demonstrated in vitro antiproliferative effects on HepG2 cells and inhibition of DHODH [46,47].

The second compound in terms of binding energy is Tipranavir, a protease inhibitor used in HIV treatment. Unlike Oteseconazole, Tipranavir has been tested on HepG2 and other cancer cell lines, where it has shown anti-proliferative effects [48,49]. Lastly, Lusutrombopag, a thrombopoietin receptor agonist, has been tested in patients with HCC to increase platelet counts [50]. However, no related in vitro studies have been conducted to explore its potential anti-proliferative effects or its ability to bind to DHODH.

For TYMS, the first compound identified was Tadalafil, a phosphodiesterase 5 inhibitor (PDE5I). Interestingly, this compound has been shown in vitro to increase the sensitivity of HCC cells to various anti-proliferative compounds [51,52]. This effect is potentially analogous to the reduction in drug resistance observed with TYMS knockdown. The second compound, Dabigatran, is a thrombin inhibitor whose derivatives have been associated with inhibitory effects on HCC and an increased efficacy of sorafenib [53,54]. Lastly, two compounds with the same binding energy were both considered. Baloxavir marboxil, a polymerase acidic endonuclease inhibitor, has been approved for treating influenza. Meanwhile, Candesartan cilexetil, an angiotensin receptor blocker, has demonstrated anticancer activity against colorectal cancer in vitro [55].

## 5. Conclusions

The metabolic flexibility of HCC, along with its heterogeneous presentation, poses challenges to the development of new and more effective systemic therapies for patients. By modeling transcriptomic data, researchers can identify or emphasize the metabolic vulnerabilities of HCC. Initial findings have highlighted potential metabolic targets within the dysregulated energy metabolism pathways of the cancer. However, the uneven distribution of targets within the OXPHOS and glycolysis pathways, coupled with limited compound data, may complicate efforts to identify drug repositioning opportunities for these pathways.

Given these challenges, attention can shift to pyrimidine metabolism, a critical pathway for HCC proliferation that offers lethal gene knockdown opportunities. The knockout of DHODH and TYMS individually could significantly impair pyrimidine metabolism, potentially leading to pyrimidine starvation, the reversal of EMT, and disruptions in the antioxidant pathways of HCC.

A drug repositioning search was conducted using publicly available IC50 data and the latest database of approved drugs in DrugBank. This analysis identified a set of compounds not currently indicated for cancer treatment. Promising inhibitors of DHODH include Oteseconazole, Tipranavir, and Lusutrombopag. For TYMS, the identified inhibitors are Tadalafil, Dabigatran, Baloxavir Marboxil, and Candesartan Cilexetil. The next step involves conducting in vitro studies to test the effectiveness of these predicted drugs on liver cancer cell lines. It is then suggested that future in vitro studies related to the proposed drugs of this paper be conducted. Furthermore, an in vitro evaluation of the simultaneous knockdown of both can shed light on the synergistic effect of the two considered genes. 

## Figures and Tables

**Figure 1 cancers-17-00903-f001:**
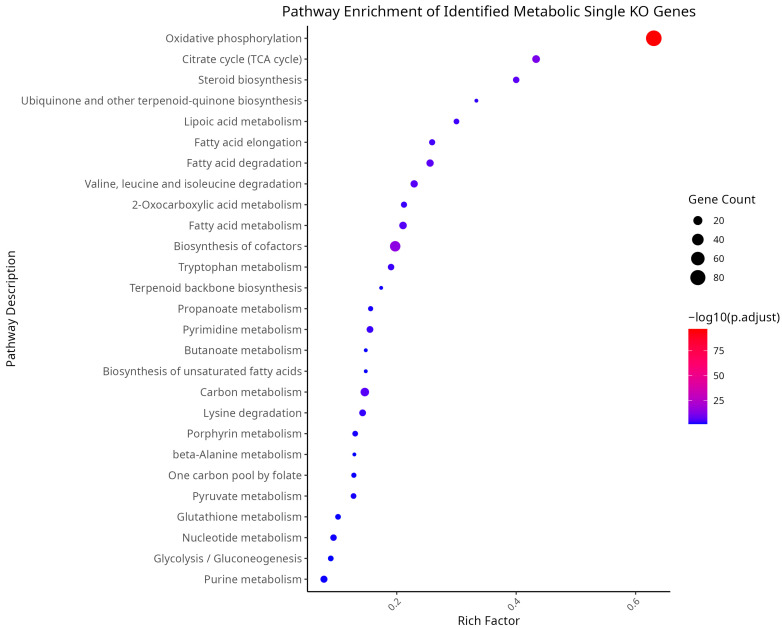
Identified metabolic biologically significant KEGG pathways of the SingleKO gene results.

**Figure 2 cancers-17-00903-f002:**
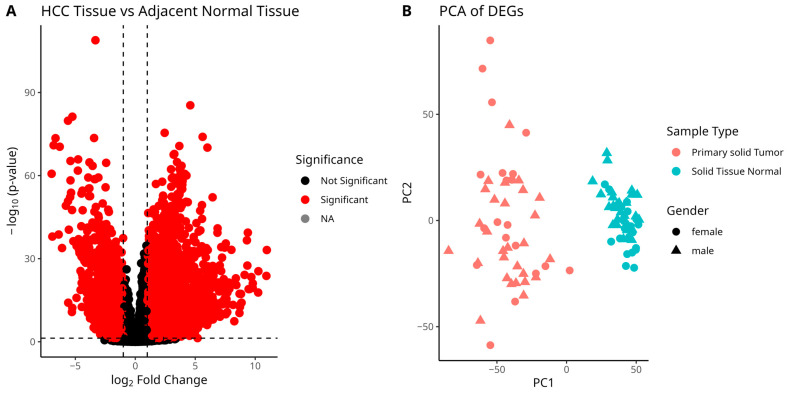
Differential gene expression analysis results of paired normal and tumor samples from the main dataset. (**A**) A volcano plot with a *p*-value threshold of 0.05 and a logFC threshold of 1. Each circle representing the gene read in Ensembl ID format. (**B**) A PCA plot of the samples based on the differentially expressed gene (DEG) list.

**Figure 3 cancers-17-00903-f003:**
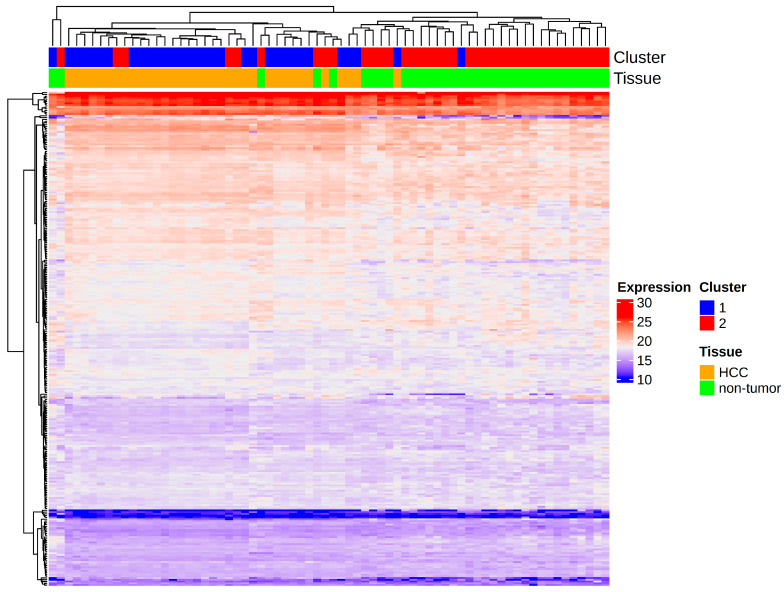
An expression heatmap of the external validation dataset depicting the results of the unsupervised clustering and the actual tissue type.

**Figure 4 cancers-17-00903-f004:**
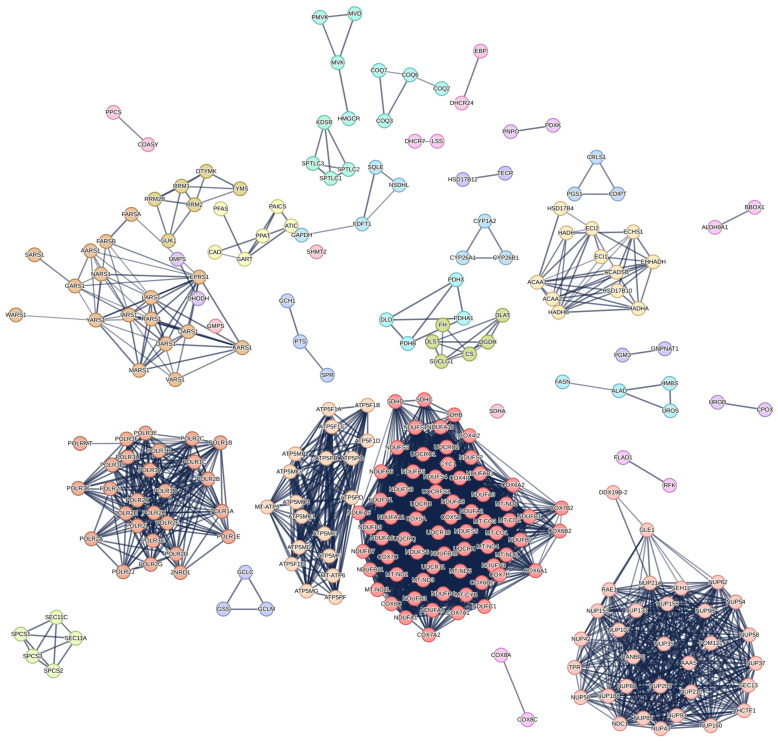
STRINGdb generated protein–protein interaction (PPI) network of the SingleKO genes. The color of the nodes depicts cluster membership.

**Figure 5 cancers-17-00903-f005:**
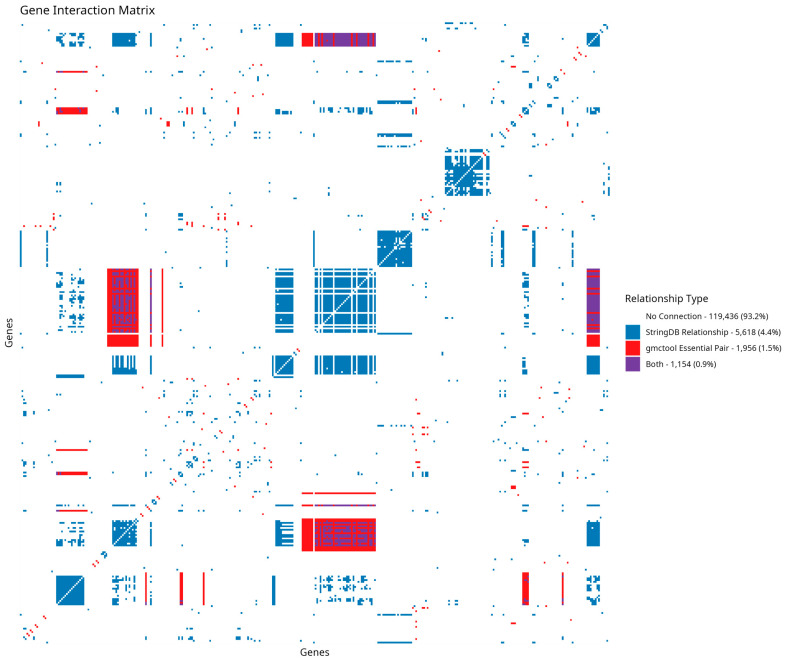
Gene interaction matrix of the genes of interest, illustrating relationships derived from STRINGdb, essential pairs identified by gmctool, and the overlap between the two. Each dot depicts a relationship between genes of interest, classified by their color.

**Figure 6 cancers-17-00903-f006:**
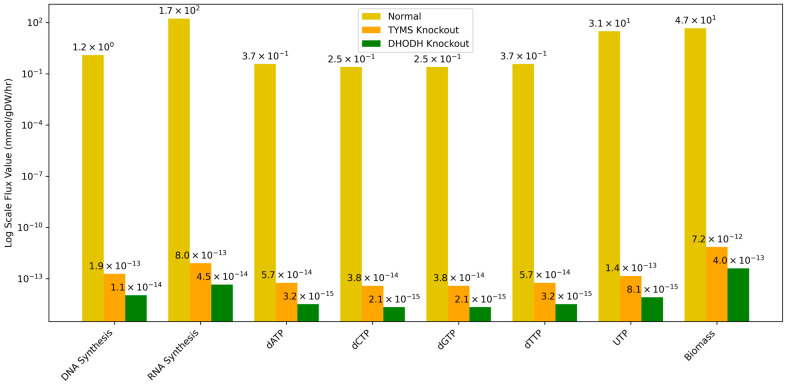
Grouped bar chart depicting the flux value of reactions relevant to the overall growth of the cell and the production of nucleotides.

**Figure 7 cancers-17-00903-f007:**
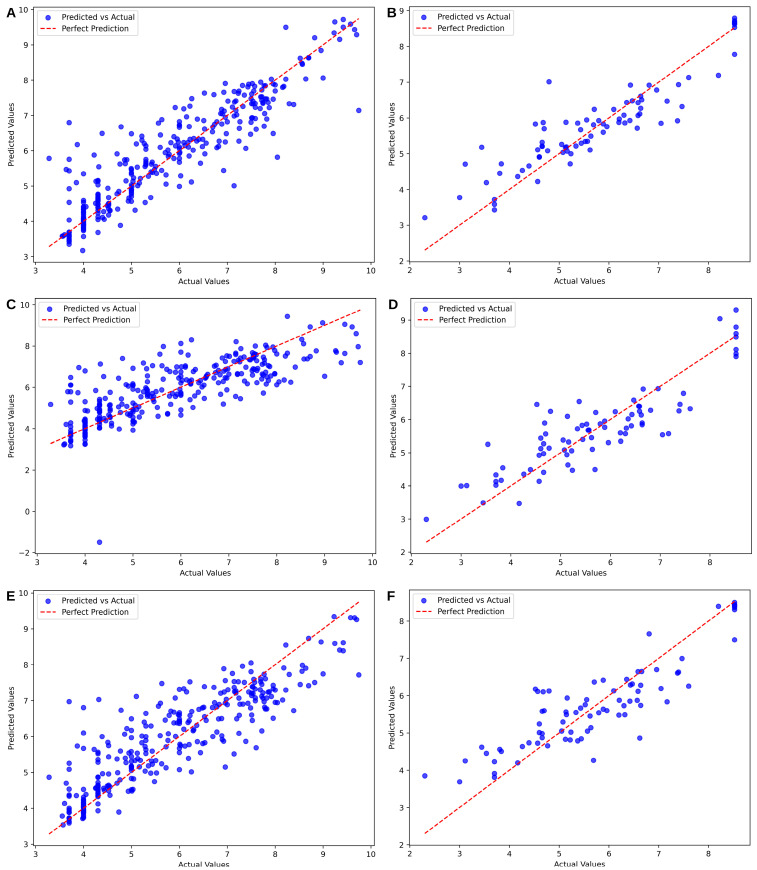
Predicted vs. actual plots for the training data of the QSAR models include: (**A**) SVM for DHODH, (**B**) SVM for TYMS, (**C**) Ridge Regression for DHODH, (**D**) Ridge Regression for TYMS, (**E**) XGBoost for DHODH, and (**F**) XGBoost for TYMS.

**Figure 8 cancers-17-00903-f008:**
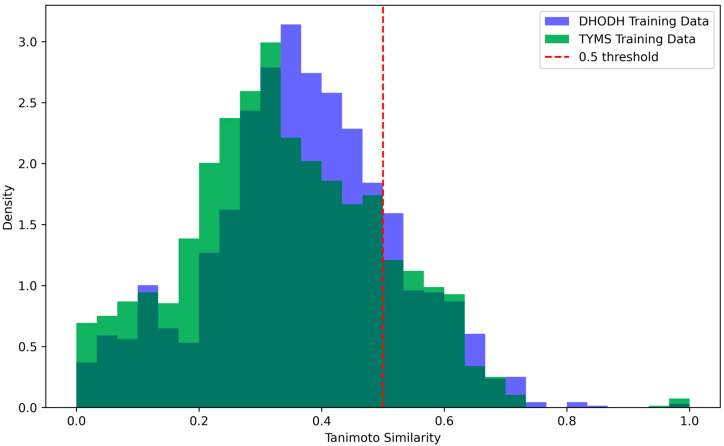
Histogram of the calculated Tanimoto similarities of approved drugs in the DrugBank database to the nearest compound in the training data.

**Figure 9 cancers-17-00903-f009:**
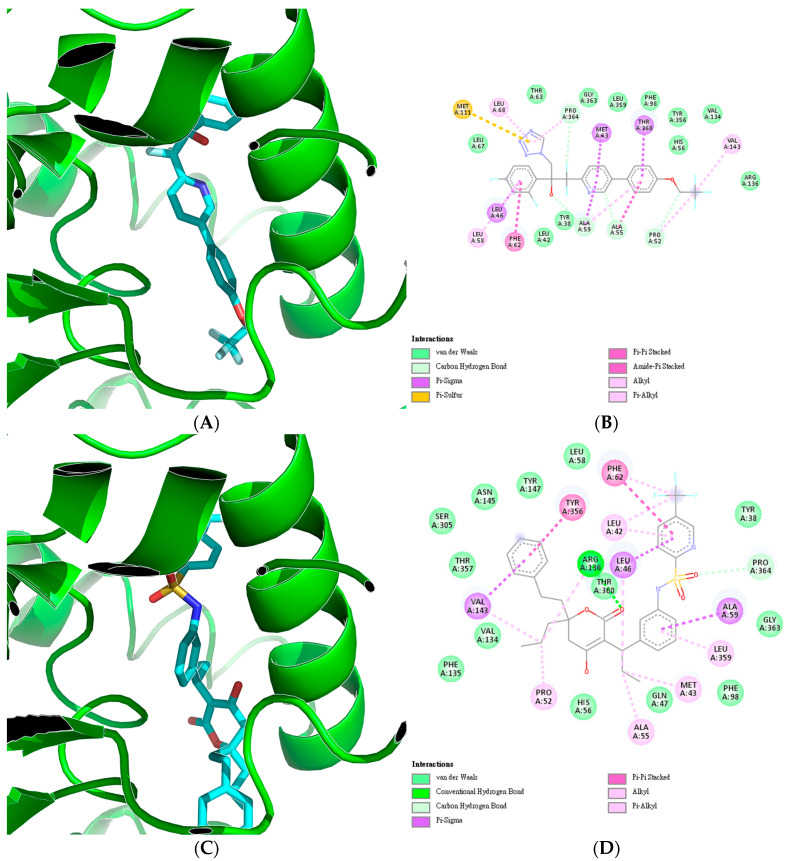
Visualization of the protein–ligand complexes and 2D interaction plots of the top three predicted drugs and a known inhibitor for DHODH, which includes (**A**) Oteseconazole-DHODH complex, (**B**) Oteseconazole-DHODH interaction plot, (**C**) Tipranavir-DHODH complex, (**D**) Tipranavir-DHODH interaction plot, (**E**) Lusutrombopag-DHODH complex, (**F**) Lusutrombopag-DHODH interaction plot, (**G**) Teriflunomide-DHODH complex, and (**H**) Teriflunomide-DHODH interaction plot.

**Figure 10 cancers-17-00903-f010:**
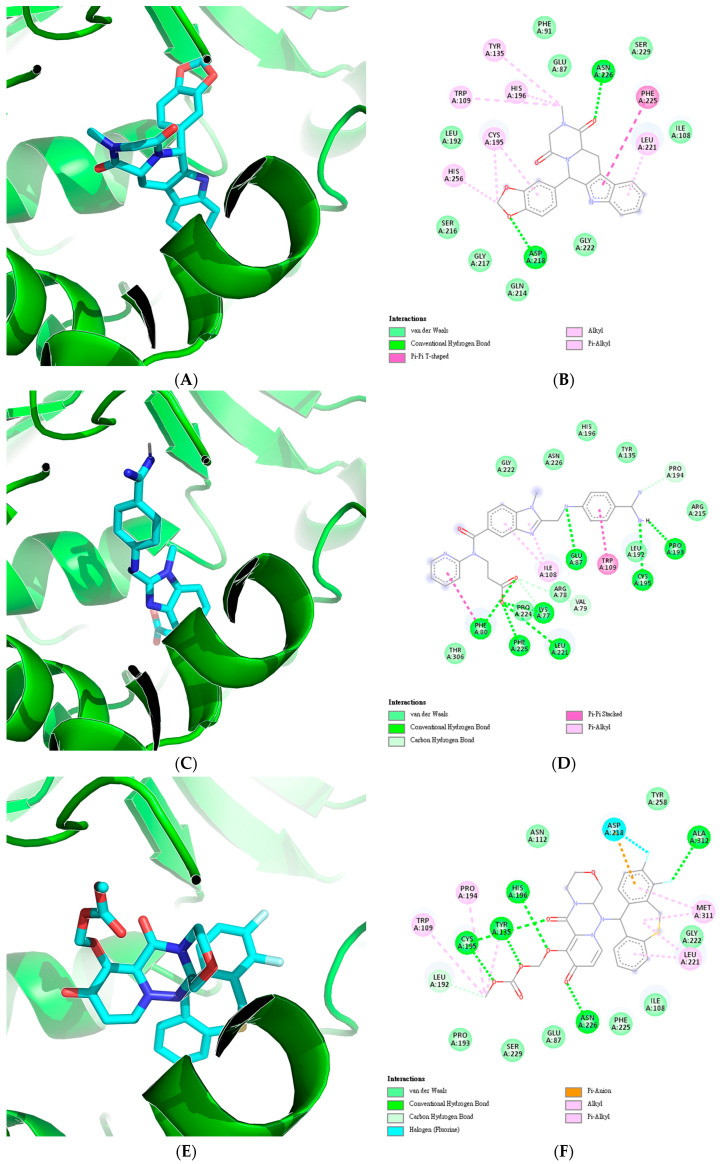
Visualization of the protein–ligand complexes and 2D interaction plots of the top four predicted drugs and a known inhibitor of TYMS which includes (**A**) Tadalafil-TYMS complex, (**B**) Tadalafil-TYMS interaction plot, (**C**) Dabigartan-TYMS complex, (**D**) Dabigartan-TYMS interaction plot, (**E**) Baloxavir marboxil-TYMS complex, (**F**) Baloxavir marboxil-TYMS interaction plot, (**G**) Candesartan cilexetil-TYMS complex, (**H**) Candesartan cilexetil-TYMS interaction plot, (**I**) Raltitrexed-TYMS complex, and (**J**) Raltitrexed-TYMS interaction plot.

**Figure 11 cancers-17-00903-f011:**
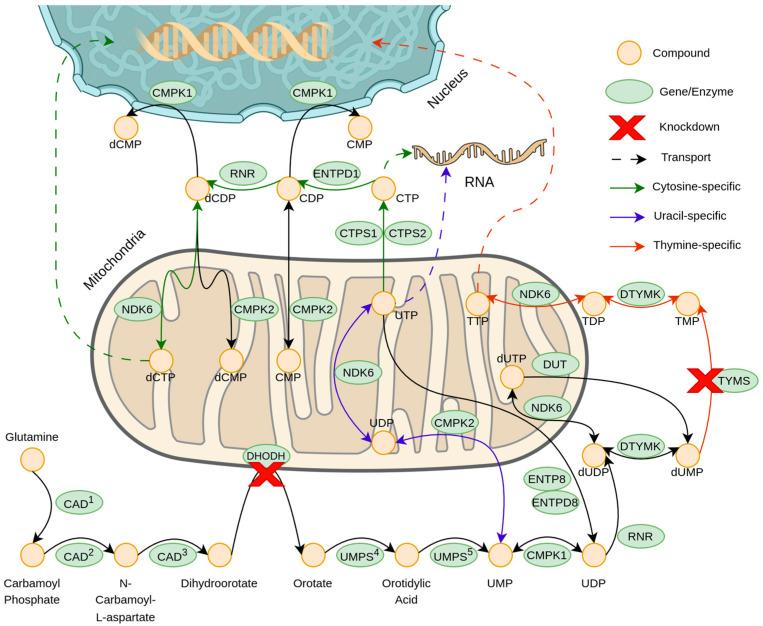
Concise de novo pyrimidine synthesis pathway with proposed knockdown locations and annotations for cytosine, uracil, and thymidine triphosphate synthesis.

**Table 1 cancers-17-00903-t001:** Flux values of reactions of interest.

Reaction	Normal (mmol/gDW/h)	TYMS Knockout (mmol/gDW/h)	DHODH Knockout (mmol/gDW/h)
DNA Synthesis	1.25 × 10^0^	1.91 × 10^−13^	1.07 × 10^−14^
RNA Synthesis	1.72 × 10^2^	8.05 × 10^−13^	4.51 × 10^−14^
dATP	3.74 × 10^−1^	5.74 × 10^−14^	3.22 × 10^−15^
dCTP	2.49 × 10^−1^	3.82 × 10^−14^	2.14 × 10^−15^
dGTP	2.49 × 10^−1^	3.82 × 10^−14^	2.14 × 10^−15^
dTTP	3.74 × 10^−1^	5.74 × 10^−14^	3.22 × 10^−15^
UTP	3.10 × 10^1^	1.45 × 10^−13^	8.12 × 10^−15^
Biomass	4.67 × 10^1^	7.16 × 10^−12^	4.02 × 10^−13^

**Table 2 cancers-17-00903-t002:** Highest quality models retrieved per algorithm and per gene with their various assessment metrics calculated from the test set.

Algorithm	Gene	Model Parameters	MAE	RMSE	R^2^	PCC	*p*-Value
SVM	DHODH	Kernel: rbf, C:10, gamma: 0.01	0.4501	0.6709	0.8210	0.9080	3.705 × 10^−119^
	TYMS	Kernel: rbf, C: 10, gamma: 0.01	0.3976	0.6306	0.8101	0.9044	1.3818 × 10^−30^
Ridge Regression	DHODH	Alpha: 1.0, Intercept: True	0.4885	0.6989	0.7667	0.8757	2.2515 × 10^−26^
	TYMS	Alpha: 0.1, Intercept: True	1.0000	1.0000	0.6022	0.7836	4.6468 × 10^−19^
XGBoost	DHODH	Estimators: 200, max depth: 5, learning rate: 0.1	0.5835	0.7639	0.7679	0.8781	2.809 × 10^−101^
	TYMS	Estimators: 200, max depth: 5, learning rate: 0.1	0.4946	0.7033	0.7638	0.8771	1.4818 × 10^−26^

**Table 3 cancers-17-00903-t003:** Top predicted drugs for DHODH inhibition.

DrugBank ID	Generic Name	Predicted pIC50	Binding Energy (kcal/mol)	Tanimoto Similarity	Approved for Cancer	Approved for HCC
DB04865	Omacetaxine mepesuccinate	8.2252	−7.6	0.67	Yes	No
DB01076	Atorvastatin	7.9997	−6	0.64	No	No
DB00275	Olmesartan	7.9988	−10.3	0.53	No	No
DB17472	Pirtobrutinib	7.9837	−13.1	0.55	Yes	No
DB09063	Ceritinib	7.9404	−9.3	0.54	Yes	No
DB00762	Irinotecan	7.8653	−10.7	0.72	Yes	No
DB15569	Sotorasib	7.8416	−6.8	0.69	Yes	No
DB00278	Argatroban	7.6664	−9.2	0.54	No	No
DB01089	Deserpidine	7.6316	−9.5	0.63	No	No
DB13055	Oteseconazole	7.5823	−12	0.59	No	No
DB01603	Meticillin	7.5724	−7.4	0.55	No	No
DB15444	Elexacaftor	7.4930	−8.2	0.67	No	No
DB08911	Trametinib	7.4893	−8.3	0.70	Yes	No
DB11431	Moxidectin	7.4381	9.5	0.54	No	No
DB12548	Sparsentan	7.3431	−7.8	0.62	No	No
DB11691	Naldemedine	7.3320	−8.2	0.76	No	No
DB00932	Tipranavir	7.3318	−11.4	0.56	No	No
DB13125	Lusutrombopag	7.3279	−11	0.56	No	No
DB15031	Daridorexant	7.3097	−8.6	0.58	No	No
DB01112	Cefuroxime	7.2844	−9.9	0.61	No	No
DB00255	Teriflunomide ^1^	6.3896	−9.8	1	Yes	No

^1^ Teriflunomide is a known DHODH inhibitor.

**Table 4 cancers-17-00903-t004:** Top predicted drugs for TYMS inhibition.

DrugBank ID	Generic Name	Predicted pIC50	Binding Energy (kcal/mol)	Tanimoto Similarity	Approved for Cancer	Approved for HCC
DB09053	Ibrutinib	8.0347	−9.9	0.55	Yes	No
DB13125	Lusutrombopag	7.8837	−7.7	0.54	No	No
DB13783	Acemetacin	7.7353	−8	0.52	No	No
DB15031	Daridorexant	7.6796	−8.3	0.58	No	No
DB09330	Osimertinib	7.6676	−7.6	0.52	Yes	No
DB15568	Adagrasib	7.5618	−9.2	0.64	Yes	No
DB00762	Irinotecan	7.5322	−10.2	0.70	Yes	No
DB09063	Ceritinib	7.5320	−8	0.51	Yes	No
DB00820	Tadalafil	7.5070	−9.9	0.61	No	No
DB00845	Clofazimine	7.5037	−8.5	0.53	No	No
DB08881	Vemurafenib	7.4767	−8.8	0.51	Yes	No
DB16390	Mobocertinib	7.4689	−7.8	0.61	Yes	No
DB00328	Indomethacin	7.4268	−8.5	0.50	No	No
DB14840	Ripretinib	7.4179	−8.9	0.54	Yes	No
DB08903	Bedaquiline	7.4054	−8.5	0.53	No	No
DB13997	Baloxavir marboxil	7.3658	−9.3	0.70	No	No
DB15149	Futibatinib	7.3568	−8.1	0.54	Yes	No
DB11656	Rebamipide	7.3454	−8.8	0.55	No	No
DB14726	Dabigatran	7.2764	−9.5	0.54	No	No
DB00796	Candesartan cilexetil	7.2675	−9.3	0.59	No	No
DB00293	Raltitrexed ^1^	6.8772	−8.4	1	Yes	No

^1^ Raltitrexed is a known TYMS inhibitor.

## Data Availability

All data used in this study were retrieved from The Cancer Genome Atlas Program (TCGA) through the use of the R package TCGAbiolinks, the ChEMBL database through its API, and the DrugBank database.

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
