# Peer review of "QSAR-Based Drug Repurposing and RNA-Seq Metabolic Networks Highlight Treatment Opportunities for Hepatocellular Carcinoma Through Pyrimidine Starvation"

_cancers, 2025, doi:10.3390/cancers17050903_

Round 1
Reviewer 1 Report
Comments and Suggestions for Authors
This is a paper carried out in silico, in which the authors address the metabolic alterations in HCC, with a view to finding new treatment opportunities in patients. I consider that, being only in silico, it should be completed with more data.
Major comments
1. The authors speak of metabolism, but in a very vague and imprecise manner, listing pathways but without providing specific data. We request the differentially expressed enzymes in tumors compared to normal adjacent liver tissues from human samples. A figure with a Volcano plot of differentially expressed enzymes and Principal component analysis of differentially expressed enzymes would be nice. This would make the results more convincing.
2. It would be nice if they could validate the metabolic signature in an external cohort of HCC transcriptomes, by Unsupervised clustering and expression heatmap of the metabolic signature score and the corresponding calculation of the metabolic score according to tissue groups.
3. Finally, I would like you to observe whether, in the case of the KO models you use, this metabolic fingerprint is altered or not.
Minor comments
The figure legends are too limited, which makes it a little difficult to understand them. I would appreciate it if the level of detail of the figure legends could be increased.
Author Response
Please see our response in the attached file.

Reviewer 2 Report
Comments and Suggestions for Authors
The manuscript is an interesting attempt to find drugs that can fight HCC by a project in silico. The tool underlying the entire study is gMCtool (gMCtool or gmctool? Use only one representation in the manuscript). It is a user-friendly web tool designed to predict essential genes crucial in the reconstruction of human metabolism. This computational tool is used to predict metabolic vulnerabilities in cancer through gMCSs (metabolic cancer signatures), using a network-based approach. It includes core modules, such as gMCS database and RNA-seq data upload. It is a powerful tool for research in metabolism and cancer.
GMCtool uses an interactome network to select genes, analyzing interactions between genes or proteins to assess prizing certain genes in metabolic pathways. It also considers gene expression data (e.g., RNA-seq.) to identify genes over- or under-expressed in specific cancer types. It assesses metabolic vulnerabilities and applies statistical methods to determine important genes based on the collected data.
The method has limitations that, if not overcome, make its conclusions uncertain. Direct validation experiments may not confirm statistically significant interactions, and differences between biological samples (e.g., HCC’s strong heterogeneity) also influence the results. However, the real limitation is the need to validate each interaction in the interactome through biophysical and biochemical experiments.
I would just like to point out that the Authors of the method used multiple myeloma as a test and in the various versions of their method, they validated the interactions that were the object of their method.
The state-of-the-art indicates that researchers have experimentally validated and declared direct interactions for only about 15% of gene or protein interactions to date. The remaining interactions are all indirect hypotheses, never validated. Therefore, interactome networks can contain a high number of artifacts, which propagate and lead to misleading results, especially when analyzing networks with high numbers of nodes. In these cases, the use of statistics can present limitations because direct experiments may not confirm statistically significant interactions.
This means that it is necessary to carry out controls at the level of the interactome network. We can perform these controls through international platforms that collect most of this information. One of them is BioGRID, which collects and curates all the physical interactions between the proteins of the human proteome, including those in many pathologies. But the Authors used STRING, which is the most complete platform for such analyses. STRING not only calculates the interactome but produces alongside it a myriad of useful and essential analyses that allow them to have complete control of their projects.
I would like to point out that this Reviewer is not contesting the project, and the methods used by the Authors but is asking for greater clarity to carry out its checks, also in order to guarantee reproducibility. After all, this should also be need the Authors themselves.
Since the Authors used STRING, they should report in the text or in the Supplements, in a clear and legible way: The complete list of proteins shown in the interactome of fig.2. The authors should clearly report the minimum parameters used to calculate the interactome (confidence score, number of nodes and edges, p-value, list of source channels used to obtain the information, any enrichments, and the number of first- and second-order proteins used), as well as the degree of each node, in the text or supplementary materials. The set of this information allows calculating the interactome to be repeated on STRING and to evaluate which interactions are at the basis of it.
For example, it is natural to think that gMCSs predicted the metabolic vulnerability of cancer cells, but HCC is a heterogeneous cancer, so the predicted genes are essential for all HCC patients?
A more critical assessment of the gMCtool results is necessary, recognizing the potential for bias and the diverse metabolic profiles across different cancers.
The approach presented is robust because it relies on predicting essential genes, but without actual evidence, it is difficult to draw reliable conclusions. All analyses produced depend on the interactome.
However, in these types of analyses, combining computational and experimental approaches is essential to get robust conclusions in cancer biology and metabolism.
Author Response

(The authors gave the same response as above.)

Round 2
Reviewer 1 Report
Comments and Suggestions for Authors
The authors have done an excellent job and have responded satisfactorily to my comments. I consider that the paper should be accepted in its present form.
Author Response
Thank you for your input on our paper.
Reviewer 2 Report
Comments and Suggestions for Authors
The authors wrote the following sentence on line 312 of their revised edition of the manuscript: “To further validate the biological significance of the main gene list, its Protein-Protein Interaction (PPI) network was generated using STRINGdb. Figure 4 illustrates the PPI network of the gene list. A highly significant PPI enrichment p-value (< 1.0E-16) was observed, indicating strong interactions among the proteins. With a high confidence score threshold of 0.9, the network contained a total of 2,424 edges, substantially exceeding the expected number of 238 edges.” These results of the STRINGdb analysis are also provided in Supplements, where I found the 278 genes of the figure 4.
This figure is the central issue of the whole analysis. The setting with a high confidence score and a filtering of the sources, operated by the authors, determined that out of 4,850 total interactions of this interactome, 87.58% of them are experimentally validated in a confidence score range between 0.700 and 0.900. This is a great result, but it brings out a serious problem. We have an interactome that starts from many single validated interactions. But this significant number of experimentally validated interactions shows us the real functional and biological validity of them. It tells us that there are numerous disconnections at various levels of organizational complexity. This prevents us from inferring which interactions are biologically relevant, not being able to consider them with respect to the other interactions in the network.
Topological analysis is based on the idea that the structure of a biological network reflects the functional relationships between its components. It measures global properties of the network, such as the centrality of nodes, modularity, connectivity degree, to infer biological roles and mechanisms of operation. However, this analysis assumes that the network is a complete and accurate representation of the underlying biological system. If the interactome has many non-connections, this assumption is violated, leading to misleading results.
Centrality measures (e.g., degree, connectivity, betweenness) identify the most "important" nodes in the network, those that potentially have a crucial role in communication and control of biological processes. However, if the interactome is incomplete, these measures can be distorted, identifying as "central" nodes that are not actually central, or vice versa. Moreover, the network is organized into modules, groups of nodes that interact intensely with each other. The analysis of modularity reveals biological pathways and specific functions. However, if the interactome is fragmented, the identified modules are incomplete or even artifacts of the lack of connections. The topological analysis is used to reconstruct biological pathways, i.e., sequences of interactions that lead to a certain result. But, if the interactome has "holes," it is difficult to reconstruct complete and accurate pathways, making it impossible to infer the functions of the proteins involved. Even if the global statistical values of the interactome (e.g., number of nodes, number of connections, degree distribution, etc.) seem good, this does not guarantee that the network is an accurate representation of the biological system. An interactome may have "normal" global statistical values but still be incomplete and distorted at the local level, i.e., in the connections between individual nodes or modules.
I'll quote just one article which should be enough: Jeong, H., Mason, S., Barabási, AL. et al. Lethality and centrality in protein networks. Nature 411, 41–42 (2001). https://doi.org/10.1038/35075138. However, there is a lot of material out there on this issue.
Fragmentation, by creating isolated groups or disconnected subgraphs, makes isolated proteins unanalyzable specifically in the context of their interactions, making it difficult to understand their biological role. Furthermore, protein complexes can be overlooked because they may not be visible due to fragmentation. To be clear, if a key interaction is represented by an isolated node, it will not be possible to determine how that protein contributes to a larger biological pathway. This means that we cannot draw meaningful conclusions from the observed interactions. In fact, the statistical techniques used to analyze networks require connectivity and a disconnected network will not meet the requirements for applying predictive models, resulting in misleading results and incorrect conclusions about protein function, despite the presence of an extremely low global p-value for the interactome. There are several reasons why this can happen: a) p-values ​​are calculated based on the probability of obtaining a certain number of observed interactions compared to a reference model (often a random model). But, even if an interactome has many disconnections, if the total number of interactions is sufficiently high, statistical sampling leads to very low p-values. This means that, statistically, the observed interactions are unlikely in a random model; b) The significance threshold for p-values ​​is often set at values ​​such as 0.05 or 0.01. Thus, even if an interactome has many disconnections, the interactions that are present may be so strong or significant that they exceed the significance threshold, leading to low p-values. However, this does not necessarily imply that the interactome is biologically significant as a whole. In the presence of many disconnections, as in this case, a low overall p-value is not always indicative of a biologically significant or optimal interactome. It is crucial to consider the overall network structure, disconnections, and biological context when interpreting statistical results.
Ultimately, without a well-connected network, where all nodes exchange direct functional relationships with each other, where any node can reach any other node in the network through the edges, it is difficult to formulate hypotheses, or predictions, about how these proteins might interact in other biological contexts. For example, if a protein is known to interact with a number of other proteins, but these interactions are not connected in a larger network, it will not be possible to predict how this protein might influence or be influenced by other cellular functions. Thus, the lack of connectivity hinders the understanding of cellular mechanisms and metabolic pathways, limiting the application of discoveries in clinical or therapeutic contexts.
I am sorry to have had to make these considerations, but they seriously affect your results and we return to the considerations I made in the first review. By then the general picture was quite clear to me but I was hoping for your recovery.
Faced with an interactome with these characteristics, you might:
- Critically evaluate the data: consider the methodological limitations and possible sources of "noise" or incompleteness of the data.
- Validate the interactions: what you have already done.
- Integrate different sources: combine the interactome with other available information (e.g., gene expression data, cellular localization data, literature data) to obtain a completer and more contextualized picture of the interactions and functions.
- Focus on modules: analyze the individual sub-graphs to identify functional modules and specific pathways, keeping in mind that they may be incomplete.
- Consider the dynamics: keep in mind that the interactome is a "snapshot" of a dynamic system, and that the interactions can vary over time and space.
I hope these suggestions are useful to revise again the manuscript, changing and improving your conclusions. The approach is interesting, but the data says otherwise. However, an attempt could still be made.
Round 3
Reviewer 2 Report
Comments and Suggestions for Authors
I appreciated your efforts, but with the improvements you have made you are now able to eliminate most of the biases because about 88% of the filtered interactions are experimentally validated. I think this is a great result for a system that has to handle one of the most heterogeneous tumors and with little reliable experimental data.
Thank you for considering my suggestions.
Good luck.